# Changes in Body Mass Index after Initiation of Antiretroviral Treatment: Differences by Class of Core Drug

**DOI:** 10.3390/v14081677

**Published:** 2022-07-29

**Authors:** Nikos Pantazis, Vasilios Papastamopoulos, Anastasia Antoniadou, Georgios Adamis, Vasilios Paparizos, Simeon Metallidis, Helen Sambatakou, Mina Psichogiou, Maria Chini, Georgios Chrysos, Periklis Panagopoulos, Nikolaos V. Sipsas, Emmanouil Barbunakis, Charalambos Gogos, Giota Touloumi

**Affiliations:** 1Department of Hygiene, Epidemiology and Medical Statistics, Medical School, National and Kapodistrian University of Athens, 11527 Athens, Greece; gtouloum@med.uoa.gr; 25th Department of Internal Medicine—Division of Infectious Diseases, Evangelismos General Hospital of Athens, 10676 Athens, Greece; vasileios.papastamopoulos@gmail.com; 34th Department of Internal Medicine, Medical School, National and Kapodistrian University of Athens, 12462 Athens, Greece; ananto@med.uoa.gr; 41st Department of Internal Medicine and Infectious Diseases Unit, General Hospital of Athens G. Gennimatas, 11527 Athens, Greece; geo.adamis@gmail.com; 5AIDS Unit, Clinic of Venereologic & Dermatologic Diseases, Medical School, National and Kapodistrian University of Athens, 16121 Athens, Greece; vpaparizos@yahoo.gr; 61st Internal Medicine Department, Medical School, Aristotle University of Thessaloniki, 54621 Thessaloniki, Greece; metallidissimeon@yahoo.gr; 7HIV Unit, 2nd Department of Internal Medicine, Medical School, Hippokration University General Hospital, 11527 Athens, Greece; helensambatakou@msn.com; 81st Department of Internal Medicine, Medical School, National and Kapodistrian University of Athens, 11527 Athens, Greece; mpsichog@yahoo.gr; 93rd Department of Internal Medicine—Infectious Diseases Unit, Red Cross General Hospital, 11526 Athens, Greece; mariachini@gmail.com; 10Infectious Diseases Unit, Tzaneion General Hospital of Piraeus, 18536 Athens, Greece; gchrysos@gmail.com; 11Department of Internal Medicine, University Hospital of Alexandroupoli, 68100 Alexandroupolis, Greece; ppanago@med.duth.gr; 12Infectious Diseases Unit, Department of Pathophysiology, Medical School, Laikon Athens General Hospital and National and Kapodistrian University of Athens, 11527 Athens, Greece; nsipsas@med.uoa.gr; 13Department of Internal Medicine, University Hospital of Heraklion, 71500 Heraklion, Crete, Greece; barbuman2003@yahoo.gr; 14Department of Internal Medicine & Infectious Diseases, Patras University General Hospital, 26504 Patras, Greece; cgogos@med.upatras.gr

**Keywords:** HIV, antiretroviral therapy, integrase strand transfer inhibitors, body mass index, weight, obesity

## Abstract

Recent research on antiretroviral treatment (ART) for HIV suggests that integrase strand transfer inhibitors (INSTIs) cause faster weight gain compared to other drug classes. Here, we investigated changes in body mass index (BMI) and obesity prevalence after treatment initiation and corresponding differences between drug classes. Data were derived from a large collaborative cohort in Greece. Included individuals were adults who started ART, in or after 2010, while previously ART naïve and achieved virologic response within the first year of ART. Data were analysed using mixed fractional polynomial models. INSTI regimens led to the more pronounced BMI increases, followed by boosted PI and NNRTI based regimens. Individuals with normal initial BMI are expected to gain 6 kg with an INSTI regimen compared to 4 kg with a boosted PI and less than 3 kg with a NNRTI regimen after four years of treatment. Prevalence of obesity was 5.7% at ART initiation and 12.2%, 14.2% and 18.1% after four years of treatment with NNRTIs, PIs, and INSTIs, respectively. Dolutegravir or Raltegravir were associated with marginally faster BMI increase compared to Elvitegravir. INSTIs are associated with faster weight gain. INSTIs’ increased risk of treatment emergent obesity and, possibly, weight-related co-morbidities should be judged against their improved efficacy and tolerability but increased clinical attention is required.

## 1. Introduction

In the early years of the HIV epidemic, weight loss was a major concern for people with HIV (PWH); weight loss was very common and had serious effects on the health of HIV positive individuals [1]. The introduction of effective antiretroviral treatment (ART) led to impressive reductions in AIDS related mortality/morbidity [2] including a substantial decrease in the incidence of HIV wasting syndrome [3]. Despite ART’s success, non-AIDS mortality and the prevalence of various comorbidities is higher among PWH compared to the general population [4]. With obesity being an established risk factor for cardiovascular disease, diabetes mellitus, chronic kidney disease and cancer, weight gain after starting ART has become a growing health concern [5].

For PWH who start ART, weight gain may be interpreted as a consequence of an overall health and appetite improvement Thus, it is sometimes considered as part of a “return-to-health” effect. However, excess weight gain, increased prevalence of obesity and high rates of related comorbidities among PWH on ART have been well established [6,7,8,9,10,11,12].

During the last five years, an increasing number of studies suggest that weight gain after treatment initiation is not uniform across the three main classes of core drugs. ART regimens based on integrase strand inhibitors (INSTI) seem to be associated with more pronounced weight gain compared to those based on non-nucleoside reverse transcriptase inhibitors (NNRTI) [13,14,15,16,17,18,19,20]. However, differences between INSTI-based regimens and those based on protease inhibitors (PI) are less clear [18,21]. Moreover, some recent studies have found that the inclusion of tenofovir alafenamide (TAF) in the NRTI (nucleoside reverse transcriptase inhibitors) backbone may have an additional effect on weight gain [19,22] although most studies focused on the effects of switching from tenofovir disoproxil fumarate (TDF) to TAF rather than starting ART with a regimen containing TAF [23,24,25,26].

In the current study, we use real world data from a multicentre cohort collaboration and focus on PWH who initiated ART while previously naive. We modelled the evolution of body mass index (BMI) rather than weight and this allowed us to estimate the prevalence of pre-obesity and obesity and their changes from baseline over time. Our analyses assessed the effects of the core drug class (i.e., INSTI vs. NNRTI vs. boosted PI) but differences between specific INSTIs and the effects of TAF were also investigated.

## 2. Materials and Methods

Data were derived from the Athens Multicenter AIDS Cohort Study (AMACS) updated at the end of 2021. AMACS is a collaborative population-based cohort study initiated in 1996 including 14 of the 16 HIV clinics in Greece. More details about AMACS are provided elsewhere [27]. The study protocol was reviewed and approved by the Hellenic Centre for Diseases Control and Prevention, the National Organization for Medicines, the Bioethics and Deontology Committee of the Medical School of the National and Kapodistrian University of Athens, and by the corresponding hospital’s scientific committee of each participating clinic.

For the current study, all adults (≥18 years old) who initiated an ART regimen based on boosted PIs, NNRTIs or INSTIs in 2010 or later, while previously ART naïve, were eligible.

We excluded individuals without weight, height, CD4 cell count or HIV-RNA viral load measurements within six months prior to ART initiation, and those who did not achieve virologic response (viral load <50 copies/mL) within the first year of ART.

Follow-up data were censored at four years after ART initiation, at the first change in the initial ART regimen or at viral rebound (first of two consecutive viral load measurements >500 copies/mL or last available above >500 copies/mL) after initial virologic response.

### 2.1. Definitions

Considered ART regimens were a combination of one boosted PI, one NNRTI or one INSTI with two NRTIs.

Classification of BMI and obesity definition was based on the following 4 WHO categories: Underweight <18.5, Normal 18.50–24.99, Preobese 25.00–29.99 and Obese ≥30.00 kg/m^2^.

### 2.2. Statistical Analysis

Baseline demographic and clinical characteristics were summarized and compared by the class of the core drug in the ART regimen using standard procedures. The evolution of BMI after ART initiation was modelled through linear mixed models after a log transformation due to its skewed distribution. Average trends over time were initially modelled (a) linearly, (b) through restricted cubic splines with three or four knots and (c) trough all possible fractional polynomials of time (degree ≤3). Based on formal and graphical assessment of all previous models, a fractional polynomial approach with two time functions (linear and square root of time) was chosen. Random terms for intercept and the two fractional polynomials of time with an unstructured covariance matrix were used in the final model.

The main covariate of interest was the ART regimen type. Alternative versions of this variable were also explored by further classification of the INSTI category: (a) including or not including tenofovir alafenamide (TAF) in the regimen’s NRTI backbone; (b) according to the actual INSTI in the regimen (Raltegravir, Dolutegravir or Elvitegravir); and (c) according to the actual INSTI in the regimen but with Raltegravir and Dolutegravir combined in one category.

Other covariates considered as potential adjusting variables included age at ART initiation, sex, transmission category, baseline CD4 cell count, baseline HIV-RNA viral load, calendar year of ART initiation, and clinical AIDS at baseline. Ethnicity was not considered as a potential covariate since 93% of the study participants were white. All covariates were tested in the model selection stage as main effects and also as interacting with the two time terms in a backwards selection procedure. The final model included only covariates with statistically significant (*p*-value < 0.05) main or interaction effects based on likelihood ratio tests. In addition, both numerical methods (based on AIC and estimates of variance components) and graphical methods were employed during the model building procedure to examine the effects of included or excluded adjusting variables on the model’s fit and predictions.

Estimated BMI evolution by ART regimen category is presented graphically for a typical combination of values for the remaining adjusting variables in the final model. Similarly, probabilities of belonging to one of the 4 BMI WHO categories were estimated (based on properties of the normal distribution and the mean and variance structure of the model) and presented graphically over time and by ART regimen category. Finally, weight changes by ART regimen category at one, two, three, and four years after ART initiation were also estimated for the same covariate pattern and assuming a height equal to the sample mean (i.e., 177 cm). These estimates are given for four values of baseline BMI (18, 23, 27, 33 kg/m^2^) equal to median sample values of participants in the four BMI WHO categories. Estimations assuming a given baseline BMI value involved methods similar to those outlined in [28].

## 3. Results

Of the 12,366 individuals in AMACS data base, 8149 adults (≥18 years old) initiated ART with a boosted PI, NNRTI or INSTI based regimen. Of these, 5323 were previously ART naive and started treatment within 2010 or later. For 3378 of them, viral load measurements within the first year of treatment were available and among them, 2590 (76.7%) achieved virologic response. Of the latter, 982 had weight, height, CD4 cell count and HIV-RNA viral load measurements available at baseline (Appendix A). Those excluded due to lack of baseline measurements (*n* = 1608) were more likely to be female (12.6% vs. 8.1%), non-MSM (36.2% vs. 32.3%), non-white (7.2% vs. 6.5%), with clinical AIDS (8.8% vs. 7.1%), and with lower median CD4 cell count (306 vs. 325 cells/μL) at baseline (Appendix A).

Demographic, clinical, and follow-up characteristics of the study sample are summarized by ART type in Table 1. The sample included 352 (35.8%) individuals who initiated ART with a boosted PI based regimen, 364 (37.1%) with a NNRTI regimen and 266 (27.1%) with an INSTI regimen. Darunavir (43.8%), Lopinavir (19.9%), and Atazanavir (19.3%) were the most common core drugs in the boosted PI group. The respective drugs in the NNRTI group were Efavirenz (59.1%) and Rilpivirine (40.9%) and in the INSTI group Elvitegravir (43.6%), Dolutegravir (28.2%) and Raltegravir (28.2%). Tenofovir Disoproxil Fumarate (TDF) with Emtricitabine was the most common backbone combination in the boosted PI (80.1%) and NNRTI (97.5%) groups whereas in the INSTI group the most common combinations were TDF with Emtricitabine (54.1%) and TAF with Emtricitabine (25.9%).

Median (IQR) age at ART initiation was 35.2 (29.7, 42.9) years with no significant differences between the three ART groups. Female sex and heterosexual or injecting drug use transmission modes were overrepresented in the boosted PI group compared to the NNRTI or INSTI groups. Additionally, individuals in the boosted PI group started ART with lower CD4 cell count, higher HIV-RNA viral load and higher proportions of pre-ART AIDS compared to the other two groups. Median year of ART initiation was 2012 in the boosted PI and INSTI groups and 2016 in the INSTI group.

Overall proportion of obesity at ART initiation was 5.0% with the corresponding median (IQR) BMI being 23.53 (21.67, 25.66) kg/m^2^ and with no significant differences between the three groups. Median time to the last BMI evaluation, included in the analyses, was shorter in the INSTI group (0.78 years) followed by the boosted PI (1.73 years) and longest in the NNRTI (2.91 years) group. Thus, the median number of analysed BMI measurements per individual followed a similar pattern (3, 5 and 6 in the INSTI, boosted PI and NNRTI groups, respectively).

Results from the final multivariable linear mixed model (Table 2) revealed that BMI levels increased significantly in all ART groups. However, rates of BMI increase differed significantly between the three ART groups (*p* = 0.014). More specifically, BMI increased faster in the INSTI group compared to the NNRTI group (*p* = 0.002) which had the slowest BMI increase. Individuals who initiated a boosted PI regimen had intermediate rates of BMI increase which did not differ significantly to those observed in the NNRTI group (*p* = 0.276). The average evolution of BMI over time, for typical values of the other covariates in the model, by ART regimen category is presented graphically in Figure 1a. As shown in this figure, rates of change in BMI are initially faster but progressively decrease. Additionally, the figure implies that the faster BMI increase in the INSTI group, leads to statistically significant differences with the NNRTI group approximately after the first year of ART, as the average trend in the INSTI group crosses the upper bound of the 95% confidence interval for the average trend in the NNRTI group.

It is also noteworthy that although estimated rates of BMI change decrease over time, they remain positive and statistically significant even at four years after ART initiation for all ART groups. However, their actual values were low, especially for the NNRTI group: the estimated (95% CI) relative (%) rates of BMI increase at four years after ART initiation were 0.65 (0.13, 1.17), 0.85 (0.26, 1.45), 1.25 (0.40, 2.11) per year for typical individuals (MSM, aged 30–39, with no AIDS and with 10,000–49,999 HIV-RNA copies/mL at baseline) in the NNRTI, boosted PI and INSTI groups, respectively.

Differences in the average evolution of BMI levels are also reflected in the proportions of obesity and preobesity after ART initiation in the three groups (Figure 1b). For example, as shown in Table 3, the estimated proportion of obesity was 5.7% in all ART groups at baseline but after four years of treatment increased to 18.1%, 14.2%, and 12.2% in the INSTI, boosted PI, and NNRTI groups, respectively, for typical individuals.

Results of the multivariable linear mixed model also revealed a significant (*p* < 0.001) correlation between the random intercept and the two-time terms implying faster BMI increases in persons with lower initial BMI. To illustrate this association, estimates of the corresponding weight gains for four typical cases in the underweight, normal, preobese and obese categories are presented in Table 4. Estimates are given by time since ART initiation and ART type with the remaining covariates assumed as in the previous examples. As shown in this table, weight gains after four years of ART are higher when initial BMI is lower. However, according to the initial BMI the magnitude of differences in weight gains is rather low and definitely much lower than the magnitude of differences between ART groups assuming a common initial BMI. For example, a typical person in the normal BMI range, is expected to gain almost 6 kg after four years of ART if treated with INSTIs compared to less than three if treated with NNRTIs.

Regarding the remaining examined covariates, only age at ART initiation, baseline HIV-RNA viral load, sex, transmission category and onset of clinical AIDS prior to ART initiation had statistically significant contribution to the main model’s fit (Table 2). Of them, only the presence of AIDS at baseline had a significant association with both initial BMI levels and their subsequent evolution whereas all of the other covariates had significant associations only with the baseline BMI values.

More specifically, persons who had already progressed to AIDS before starting ART had approximately 1 kg/m^2^ (95% CI: 0.04, 2.08) lower initial BMI, but their BMI levels increased much faster while on ART; after four years of ART, they had on average approximately 2 kg/m^2^ (95% CI: 0.43, 3.93) higher BMI than those who were AIDS free at baseline.

Finally younger age, higher baseline HIV-RNA viral load, female sex, and infection through injecting drug use were associated with lower initial BMI but had no statistically significant effects on BMI trends after ART initiation.

### Sensitivity Analyses

Similar modelling, with further stratification of the INSTI group according to the presence or absence of TAF in the NRTI backbone, revealed faster BMI increases, especially in the early phase of ART, when INSTIs were combined with a TAF based NRTI backbone (Figure 2a). However, the specific ART regimen grouping did not improve significantly the fit of the main model presented earlier (*p* = 0.822).

Stratification of the INSTI group according to the core drug (Elvitegravir, Dolutegravir or Raltegravir) revealed that Elvitegravir was associated with milder BMI increases compared to both Dolutegravir or Raltegravir. Combining the two latter drugs into one INSTI category and assigning Elvitegravir regimens in a separate INSTI category, marginally (*p* = 0.056) improved the fit of the main model. The results of this last model are presented graphically in Figure 2b.

## 4. Discussion

In this study, we used repeated measurements of weight, taken at ART initiation and up to four years later, from PWH who started ART with a NNRTI, boosted PI, or INSTI based regimen while previously ART-naive. Data were derived from a collaboration of the largest clinics treating PWH in Greece. The multivariable analysis of the resulting BMI values showed that, adjusting for potential confounders, the class of the core drug in the first ART regimen is associated with significantly different BMI evolution while on ART. PWH starting an INSTI regimen had the more pronounced BMI increases, followed by those receiving a boosted PI regimen and those receiving a NNRTI based regimen. Typical individuals with a normal initial BMI are expected to gain approximately 6 kg of body weight with an INSTI regimen compared to 4 kg with a boosted PI and less than 3 kg with a NNRTI based regimen after four years of treatment. Consequently, the prevalence of obesity is expected to rise from 5.7% at ART initiation to 12.2%, 14.2%, and 18.1% after four years of treatment with NNRTIs, PIs and INSTIs, respectively. Additionally, there were indications that there are differences within the INSTI class with individuals receiving a Dolutegravir or Raltegravir containing regimen, having steeper BMI increases compared to those on Elvitegravir. The inclusion of TAF in the backbone of an INSTI based regimen seemed to be associated with a transient and faster initial BMI increase compared to INSTI regimens without TAF, but differences were not statistically significant.

Our results are consistent with those from most other studies, which investigated the same issue in other populations. Faster weight or BMI increases when using INSTIs in first line ART regimens have been demonstrated in both clinical trials [13,20,29,30] and observational studies [14,15,16,17,18,31] in North America, Europe, South America and Africa. Sax and colleagues [19] meta-analysed data from eight randomized clinical trials and found that INSTIs were associated with more weight gain compared to PIs or NNRTIs.

Using data from a large collaboration of cohorts in Europe and Australia, Bansi-Matharu and colleagues [32] found that Dolutegravir, Raltegravir, and TAF were independently associated with an increase of at least 7% in BMI compared to pre-ART levels. TAF effects on weight gain have been reported in many other studies [19,20,22]. In our study, the number of participants starting ART with a TAF containing regimen was relatively small thus, although we noticed an initial faster BMI increase with regimens containing an INSTI and TAF, the difference with INSTI regimens without TAF was not statistically significant.

Similarly to other studies, we found indications that Dolutegravir and Raltegravir were associated with more weight gain compared to Elvitegravir. A recently published network meta-analysis [33] estimated that Dolutegravir, followed closely by Bictegravir and then Raltegravir, are associated with more weight gain whereas Elvitegravir was associated with the lowest increase. Dolutegravir and Bictegravir were also found to be associated with more weight gain compared to NNRTIs and PIs in a meta-analysis of eight clinical trials [19] and significantly more weight gain was observed with Dolutegravir compared to Elvitegravir in a retrospective, observational cohort study [17].

Regarding other prognostic factors, we found that individuals who had already progressed to AIDS before initiating ART had faster BMI increase. This is consistent with results from other studies [19,31]. However, other studies have reported similar effects for low CD4 cell count and/or high HIV-RNA viral load instead or on top of AIDS at baseline [16,18,22,31,32]. In our study, baseline AIDS was a better predictor for BMI gain whereas baseline CD4 cell count and HIV-RNA viral load had no additional statistically significant effect on the rate of BMI increase. In general, AIDS and/or low CD4 cell count at baseline are associated with more advanced disease and excessive weight gains may reflect either a “return-to-health” effect or, in some cases, overcompensation for poor pre-ART health status.

In our study, female sex although associated with lower initial BMI, had no statistically significant effect on BMI trends after ART initiation. Female sex is reported by several studies as a risk factor for weight gain, especially among those on INSTI based regimens [14,16,34]. However, and similarly to our findings, a recently published large multicohort study found no significant association between sex and weight gain [32]. Unfortunately, we were not able to investigate potential ethnicity effects as the vast majority of study participants in our study were white.

In general, the lack of significant effects of low CD4 cell count, high HIV-RNA viral load, and female sex on BMI trends after ART initiation could be due to lack of power or due to the adjustment for AIDS at baseline in our model. Additionally, these effects could be mediated through lower initial BMI, which in our analysis is modelled by allowing correlations between the random intercept and random BMI time trends effects. Indeed, in our analysis we found significant associations between baseline levels and subsequent BMI trends with individuals with lower baseline BMI having faster increases while on ART.

It is also noteworthy that, given the aforementioned factors with statistically significant effects, the calendar year of ART initiation was not associated with baseline BMI nor with its rate of change after ART initiation. Thus, confounding effects on the association between ART regimen class and rate of BMI increase seem unlikely.

The main limitation of our study is its moderate sample size and the relatively short follow-up time especially for participants on INSTI regimens. However, we tried to minimize the effects of the later introduction of INSTIs in clinical practice by restricting the analysis to those who started ART on 2010 or later and by artificially censoring follow-up at four years after ART initiation.

Additional limitations included the lack of detailed data on adherence to treatment, the low percentages of non-white and female participants, the relatively low number of those receiving INSTI regimens containing TAF, and the limited variability in NRTI backbone combinations. The NNRTI and boosted PI groups included individuals who received relatively old antiretrovirals (i.e., Efavirenz, Lopinavir, and Atazanavir) while the latter group was also characterized by a higher proportion of women compared to the other two groups. Regarding the lack of adherence data, we only included individuals who achieved virologic response and censored their data in case of a virologic rebound in an effort to analyse data under a proxy of treatment adherence. The low proportion of non-white participants did not allow us to investigate potential ethnicity effects. The limited variability in NRTI backbone combinations (e.g., TDF and Emtricitabine was the backbone in the vast majority of participants in the boosted PI and NNRTI groups) precluded us from a more detailed investigation of their effects independently of or in combination with the main drug in ART regimens.

Finally, the exclusion of individuals without weight, height, CD4 cell count and viral load measurements at baseline may limit the generalizability of our findings. The excluded population is mainly characterized by a higher proportion of women (12.6% vs. 8.2%) and a higher proportion of individuals with more advanced disease (8.8% vs. 4.3% with a previous AIDS diagnosis). Although the magnitude of differences in demographic and clinical characteristics between excluded and included individuals is relatively small, our results should be interpreted with caution especially when generalized to populations with different characteristics compared to those of the study sample. It should be noted that exclusions due to missing viral load measurements in earlier steps of the sample selection procedure are more likely to be completely at random as the sparseness of such measurements was caused by administrative issues which affected the availability of reagents for viral load quantification in the whole country during a substantial portion of the study period.

The main strength of our study is the formal statistical analysis of all available data using a flexible multivariable mixed model which was carefully selected among a wide range of alternative models. This allowed us to estimate BMI trends according to various baseline characteristics and types of treatment but also to derive estimates of obesity and preobesity prevalence continuously over time (without having to rely on the availability of BMI measurements at specific time points). Additionally, the main model allowed us to quantify the effects of initial BMI levels on the corresponding subsequent trends. These estimates revealed that ISNTIs are associated with faster BMI increase and for typical PWH may lead to a substantial increase in the prevalence of obesity. Additionally, our estimates suggest that significant weight gain, when treated with INSTIs, is to be expected not only for individuals who are underweight at baseline but also for those in or above the normal range.

Given that individuals in our study, and especially those who received INSTIs, were in a relatively good health status (i.e., median CD4 358 cells/μ, median BMI 23.7 kg/m^2^) at ART initiation, the BMI increase in the INSTI group seems to exceed what would be expected from a “return-to-health” effect. The increase in the probability of obesity raises concerns regarding potential health impact through increased risk for cardiovascular disease, metabolic or renal disorders and cancers.

Our results reinforce the notion that INSTIs are associated with substantial weight gain, especially when combined with TAF or when they include Dolutegravir or Raltegravir rather than Elvitegravir. Further clinical research with currently used antiretrovirals is required to accurately quantify the corresponding drug specific effects, determine the underlying biological mechanisms, and assess the potential health impact. Risks related to obesity should be carefully judged against the improved tolerability and potency of modern ART but, at the same time, increased clinical attention and counseling is required to help PWH on treatment to maintain a healthy body weight.

## Figures and Tables

**Figure 1 viruses-14-01677-f001:**
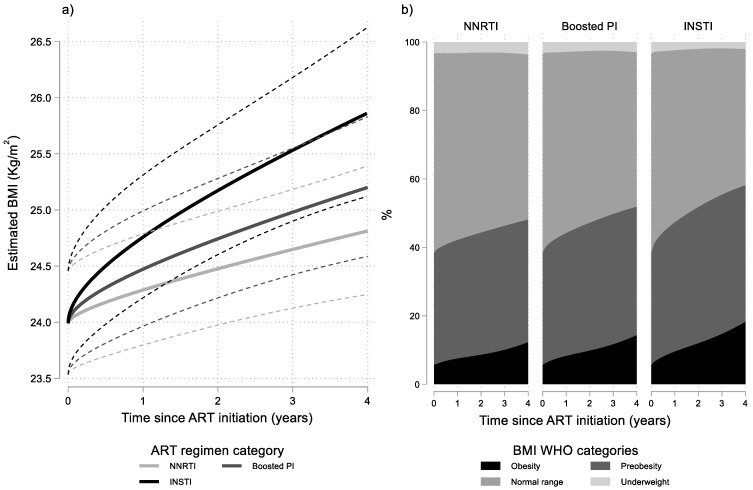
Results from multivariable linear mixed model: (**a**) estimated BMI (95% CI) after cART initiation by ART regimen category and baseline BMI; and (**b**) estimated probabilities for BMI WHO categories by time since ART initiation and ART regimen category. Estimates shown for men having sex with men (MSM), aged 30–39, with no AIDS and with 10,000–49,999 HIV-RNA copies/mL at baseline.

**Figure 2 viruses-14-01677-f002:**
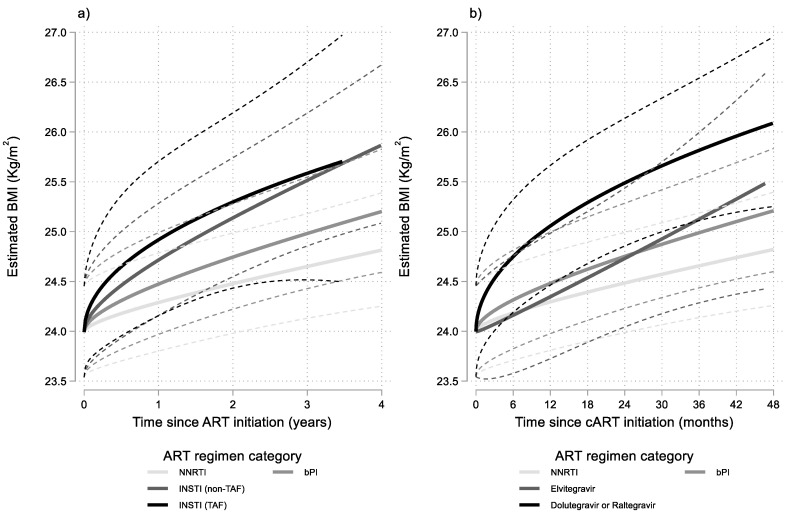
Results from multivariable linear mixed model: estimated BMI (95% CI) after cART initiation by time since ART initiation and ART regimen category. Estimates shown for men having sex with men (MSM), aged 30–39, with no AIDS and with 10,000–49,999 HIV-RNA copies/mL at baseline. INSTI regimens stratified into: (**a**) containing or not containing TAF and (**b**) using Elvitegravir or using Dolutegravir or Raltegravir.

**Table 1 viruses-14-01677-t001:** Demographic, clinical and follow-up characteristics of the study sample (*n* = 982) by ART regimen category. All figures are mdian (IQR) or *n* (%).

	ART Regimen Category			Overall 982 (100.0%)	*p*-Value
	Boosted PI ^1^ 352 (35.85%)	NNRTI ^2^ 364 (37.07%)	INSTI ^3^ 266 (27.09%)		
Age at ART initiation (years)	35.6 (29.9, 43.6)	34.2 (29.8, 41.3)	35.4 (29.3, 44.1)	35.2 (29.7, 42.9)	0.194
Female sex	38 (10.80%)	25 (6.87%)	17 (6.39%)	80 (8.15%)	0.075
Transmission mode					<0.001
*MSM* ^4^	192 (54.55%)	259 (71.15%)	214 (80.45%)	665 (67.72%)	
*PWID* ^5^	70 (19.89%)	37 (10.16%)	22 (8.27%)	129 (13.14%)	
*Heterosexual*	71 (20.17%)	48 (13.19%)	25 (9.40%)	144 (14.66%)	
*NA* ^6^	19 (5.40%)	20 (5.49%)	5 (1.88%)	44 (4.48%)	
Ethnic/racial group					0.015
*White*	320 (90.91%)	339 (93.13%)	259 (97.37%)	918 (93.48%)	
*Black*	3 (0.85%)	4 (1.10%)	2 (0.75%)	9 (0.92%)	
*Asian*	0 (0.00%)	2 (0.55%)	1 (0.38%)	3 (0.31%)	
*NA* ^6^	29 (8.24%)	19 (5.22%)	4 (1.50%)	52 (5.30%)	
Baseline ^7^ CD4 (cells/μL)	263 (128, 376)	342 (269, 434)	358 (249, 499)	325 (212, 428)	<0.001
Baseline ^7^ HIV-RNA (log_10_ copies/mL)	4.59 (4.00, 5.09)	4.45 (3.95, 4.82)	4.46 (3.71, 5.03)	4.50 (3.94, 4.99)	0.022
Year of ART initiation	2012 (2011, 2013)	2012 (2011, 2013)	2016 (2014, 2018)	2013 (2011, 2015)	<0.001
Months from diagnosis to ART	2.71 (0.82, 7.82)	9.36 (2.68, 28.45)	1.58 (0.62, 4.47)	3.45 (1.12, 13.11)	<0.001
ART NRTI ^2^ backbone					<0.001
*Tenofovir disoproxil* *fumarate & Emtricitabine*	282 (80.11%)	355 (97.53%)	144 (54.14%)	781 (79.53%)	
*Tenofovir alafenamide &* *Emtricitabine*	0 (0.00%)	3 (0.82%)	69 (25.94%)	72 (7.33%)	
*Lamivudine & Abacavir*	64 (18.18%)	6 (1.65%)	43 (16.17%)	113 (11.51%)	
*Other*	6 (1.70%)	0 (0.00%)	10 (3.76%)	16 (1.63%)	
AIDS before baseline	22 (6.25%)	12 (3.30%)	8 (3.01%)	42 (4.28%)	0.072
Baseline ^7^ Weight (kg)	72.0 (65.0, 80.0)	74.5 (68.0, 83.0)	75.0 (67.0, 82.0)	74.0 (66.0, 82.0)	0.005
Baseline ^7^ Height (cm)	176 (171, 180)	178 (173, 183)	178 (172, 182)	1.78 (1.72, 1.82)	0.002
Baseline ^7^ BMI ^8^ (kg/m ^2^)	23.36 (21.47, 25.13)	23.66 (21.69, 25.85)	23.67 (21.71, 25.93)	23.53 (21.67, 25.66)	0.182
Baseline ^7^ BMI ^8^ classification					0.293
Underweight	19 (5.40%)	10 (2.75%)	11 (4.14%)	40 (4.07%)	
Normal range	237 (67.33%)	238 (65.38%)	167 (62.78%)	642 (65.38%)	
Preobesity	83 (23.58%)	93 (25.55%)	75 (28.20%)	251 (25.56%)	
Obesity	13 (3.69%)	23 (6.32%)	13 (4.89%)	49 (4.99%)	
Follow-up time (years)	1.73 (0.09, 3.44)	2.91 (0.54, 3.68)	0.78 (0.00, 2.43)	1.80 (0.02, 3.47)	<0.001
Number of BMI measurements	5.0 (2.0, 9.0)	6.0 (2.0, 9.0)	3.0 (1.0, 5.0)	4.0 (2.0, 8.0)	<0.001

^1^ Protease inhibitors; ^2^ non-nucleoside reverse transcriptase inhibitors; ^3^ integrase strand inhibitors; ^4^ men having sex with men; ^5^ people who inject drugs; ^6^ non-available; ^7^ at ART initiation; ^8^ body mass index.

**Table 2 viruses-14-01677-t002:** Results from the final multivariable linear mixed model for the evolution of body mass index (log transformed) after ART initiation.

Covariate	Coefficient	95% CI	*p*-Value (df ^1^)
**Intercept (reference category ^2^)**	3.178	(3.158, 3.197)	<0.001 (1)
**Time (in years) trend (reference category ^2^)**			<0.001 (1)
Time-1: Square root of time	0.008	(−0.008, 0.023)	
Time-2: Time	0.005	(−0.004, 0.013)	
**Interaction**: ART regimen category on time trend			0.014 (4)
ART regimen category X Time-1			
*Boosted PI* ^3^ */NNRTI* ^4^	0.007	(−0.015, 0.030)	
*INSTI* ^5^/NNRTI ^4^	0.018	(−0.009, 0.045)	
ART regimen category X Time-2			
*Boosted PI* ^3^ */NNRTI* ^4^	0.000	(−0.012, 0.013)	
*INSTI* ^5^/NNRTI ^4^	0.002	(−0.014, 0.017)	
**Main effect**: Age (years)			<0.001 (3)
*20–29/30–39*	−0.059	(−0.081, −0.037)	
*40–49/30–39*	0.035	(0.012, 0.058)	
*50+/30–39*	0.010	(−0.023, 0.043)	
**Main effect**: HIV-RNA at ART initiation (copies/mL)			0.010 (4)
*<500/10,000–49,999*	0.040	(−0.001, 0.080)	
*500–9999*/*10,000–49,999*	0.010	(−0.014, 0.035)	
*50,000–99,999*/*10,000–49,999*	−0.008	(−0.034, 0.019)	
*100,000+*/*10,000–49,999*	−0.025	(−0.049, −0.002)	
**Main effect**: AIDS before ART initiation			0.047 (1)
*Yes/No*	−0.045	(−0.090, −0.001)	
**Interaction**: AIDS before ART initiation on time trend			<0.001 (2)
AIDS before ART X Time-1: *Yes/No*	0.178	(0.123, 0.233)	
AIDS before ART X Time-2: *Yes/No*	−0.057	(−0.088, −0.025)	
**Main effect**: Sex &Transmission category			<0.001 (5)
*Male PWID* ^6^ */MSM* ^7^	−0.066	(−0.094, −0.038)	
*Female PWID* ^6^*/MSM* ^1^	−0.100	(−0.171, −0.028)	
*Male Heterosexual/MSM* ^7^	0.027	(−0.005,−0.060)	
*Female Heterosexual/MSM* ^7^	0.004	(−0.033, 0.042)	
*NA* ^8^*/MSM* ^7^	0.057	(0.013, 0.100)	

^1^ Degrees of freedom of Wald test (global test if df > 1); ^2^ MSM, aged 30–39, with no AIDS and with 10,000–49,999 HIV-RNA copies/mL at baseline; ^3^ protease inhibitors; ^4^ non-nucleoside reverse transcriptase inhibitors; ^5^ integrase strand inhibitors; ^6^ people who inject drugs; ^7^ men having sex with men; ^8^ non-available.

**Table 3 viruses-14-01677-t003:** Estimated probabilities of belonging to one of the body mass index WHO categories by time since ART initiation and ART regimen category. Estimates shown for men having sex with men, aged 30–39, with no AIDS and with 10,000–49,999 HIV-RNA copies/mL at baseline.

Time since ART Initiation (Years)	ART Regimen Category	Estimated Probabilities (%)			
		Underweight	Normal Range	Preobesity	Obesity
	NNRTI ^1^	3.3	58.1	32.8	5.7
0 (Baseline)	Boosted PI ^2^	3.3	58.1	32.8	5.7
	INSTI ^3^	3.3	58.1	32.8	5.7
	NNRTI ^1^	3.2	54.7	34.7	7.5
1	Boosted PI	2.8	53.0	36.0	8.2
	INSTI ^3^	2.3	50.3	37.9	9.5
	NNRTI ^1^	3.0	52.7	35.8	8.5
2	Boosted PI ^2^	2.5	50.3	37.5	9.7
	INSTI ^3^	1.9	46.2	40.0	11.9
	NNRTI ^1^	3.0	50.7	36.4	9.9
3	Boosted PI ^2^	2.5	47.7	38.2	11.6
	INSTI ^3^	1.8	42.8	40.9	14.6
	NNRTI ^1^	3.6	48.3	36.0	12.2
4	Boosted PI ^2^	2.9	45.2	37.7	14.2
	INSTI ^3^	2.0	39.8	40.2	18.1

^1^ Non-nucleoside reverse transcriptase inhibitors; ^2^ protease inhibitors; ^3^ integrase strand inhibitors.

**Table 4 viruses-14-01677-t004:** Estimated (95% CI) weight gains (kg) by time since ART initiation, baseline Body Mass Index (BMI) and ART regimen category. Estimates shown for men having sex with men, aged 30–39, with no AIDS and with 10,000–49,999 HIV-RNA copies/mL at baseline and assuming height equals 177 cm.

Baseline BMI (kg/m^2^)	ART Regimen Category	Time since ART Initiation			
		1 Year	2 Years	3 Years	4 Years
	NNRTI ^1^	1.79 (0.95, 2.63)	2.48 (1.52, 3.43)	2.99 (1.85, 4.13)	3.41 (1.95, 4.86)
18	Boosted PI ^2^	2.23 (1.36, 3.10)	3.12 (2.12, 4.12)	3.79 (2.58, 4.99)	4.34 (2.79, 5.89)
	INSTI ^3^	2.92 (1.94, 3.89)	4.15 (3.02, 5.28)	5.12 (3.71, 6.52)	5.94 (4.07, 7.80)
	NNRTI ^1^	1.09 (0.46, 1.73)	1.71 (0.99, 2.42)	2.25 (1.41, 3.08)	2.74 (1.69, 3.80)
23	Boosted PI ^2^	1.65 (0.96, 2.34)	2.51 (1.72, 3.30)	3.24 (2.29, 4.19)	3.91 (2.69, 5.14)
	INSTI ^3^	2.51 (1.63, 3.39)	3.81 (2.79, 4.83)	4.91 (3.62, 6.20)	5.91 (4.17, 7.65)
	NNRTI ^1^	NS ^4^	NS ^4^	1.45 (0.35, 2.55)	2.01 (0.62, 3.40)
27	Boosted PI ^2^	1.03 (0.15, 1.92)	1.83 (0.82, 2.85)	2.61 (1.39, 3.82)	3.36 (1.79, 4.94)
	INSTI ^3^	2.03 (0.94, 3.12)	3.34 (2.08, 4.60)	4.53 (2.94, 6.12)	5.67 (3.54, 7.81)
	NNRTI ^1^	NS ^4^	NS ^4^	NS ^4^	NS ^4^
33	Boosted PI ^2^	NS ^4^	NS ^4^	NS ^4^	NS ^4^
	INSTI ^3^	NS ^4^	2.37 (0.28, 4.46)	3.68 (1.11, 6.26)	5.03 (1.63, 8.44)

^1^ Non-nucleoside reverse transcriptase inhibitors; ^2^ protease inhibitors; ^3^ integrase strand inhibitors; ^4^ statistically non-significant; baseline BMI values correspond to median sample values of participants in the four BMI WHO categories.

## Data Availability

AMACS individual-level data are derived from collaborating HIV clinics and although individual data do not include patient names or identifying information of the participants, as data contain potentially sensitive information, there are ethical restrictions imposed by the Bioethics and Deontology Committee of the Medical School of the National and Kapodistrian University of Athens. Anonymized individual data can be shared after interested researchers submit a concept sheet to the AMACS steering committee (chair: Giota Touloumi, email: gtouloum@med.uoa.gr) and the Bioethics and Deontology Committee of the Medical School of the National and Kapodistrian University of Athens (bioethics@med.uoa.gr).

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
