# Peer review of "Changes in Body Mass Index after Initiation of Antiretroviral Treatment: Differences by Class of Core Drug"

_viruses, 2022, doi:10.3390/v14081677_

Round 1

Reviewer 1 Report

Knowing that weight gain after starting ART is common, and that INSTI based regimens are associated with more pronounced weight gains compared to NNRTI, the authors sought to elucidate differences in BMI change between INSTI based and boosted PI based regimens.

While I think this is an important question and that the authors used rigorous statistical methods, I have several concerns about epidemiologic methods.

My biggest concern is the lack of overlap in calendar time between regimens. The median year for PI and NNRTI was 2012 versus 2016 for INSTI. Only 25% of PI or NNRTI regimens were initiated after 2013. Only 25% of INSTI regimens were initiated before 2014. We know that obesity is increasing over time. Could it be that people initiating ART in 2016 would gain more weight than those who initiated in 2012, no matter what regimen they used? With enough overlap, adjustment for calendar year might be possible. Without adequate overlap, intractable confounding and sparse data bias are a concern. (Greenland et al, Sparse data bias: a problem hiding in plain sight, BMJ 2016)

 The authors did not adjust for calendar year. Their model selection was based only on p-value of main or interaction effects. This is a naïve approach that doesn’t consider change in estimate of the association of interest or model fit.

Finally, I’m concerned about potential selection bias and generalizability. Only 63% of 5,323 ART initiators had viral load at 1 year. Only 49% of initial sample achieved viral suppression. Only 982 (18%) had complete baseline information. Does the proportion included in the final sample differ by regimen or calendar year. How representative are the 982 of the full sample?

Author Response

Knowing that weight gain after starting ART is common, and that INSTI based regimens are associated with more pronounced weight gains compared to NNRTI, the authors sought to elucidate differences in BMI change between INSTI based and boosted PI based regimens.

While I think this is an important question and that the authors used rigorous statistical methods, I have several concerns about epidemiologic methods.

1) My biggest concern is the lack of overlap in calendar time between regimens. The median year for PI and NNRTI was 2012 versus 2016 for INSTI. Only 25% of PI or NNRTI regimens were initiated after 2013. Only 25% of INSTI regimens were initiated before 2014. We know that obesity is increasing over time. Could it be that people initiating ART in 2016 would gain more weight than those who initiated in 2012, no matter what regimen they used? With enough overlap, adjustment for calendar year might be possible. Without adequate overlap, intractable confounding and sparse data bias are a concern. (Greenland et al, Sparse data bias: a problem hiding in plain sight, BMJ 2016). The authors did not adjust for calendar year.

We thank the reviewer for this very reasonable comment. There is obviously an association between calendar year and type of ART regimen, driven mainly by the continuous introduction of new drugs and the subsequent changes in treatment guidelines. However, there was enough overlap between the three main drug classes which allowed us to test for any calendar year effects in both baseline BMI and its rate of change. The vast majority of participants in the PI and NNRTI group initiated ART up to 2015 (i.e. 2015 was the 95th percentile of calendar year of ART initiation for these two groups). At the same time though, 127 of the 266 participants in the INSTI group (47.7%) initiated treatment before or in 2015.

As we note in the statistical methods section of the original version of our manuscript, calendar year of ART initiation was one of the potential confounders assessed during the model building stage. The specific covariate was not included in the final multivariable model as its effects were clearly non significant (p-value for effect on baseline 0.953, p-value for interaction effects with the first and second time variable 0.558 and 0.937, respectively) and practically were not affecting the effect sizes of other covariates in the model.

However we agree that this point requires more details thus we briefly mention it in the discussion section of the revised manuscript (page 23, 2nd paragraph).

2) Their model selection was based only on p-value of main or interaction effects. This is a naïve approach that doesn’t consider change in estimate of the association of interest or model fit.

Our model building procedure followed an approach similar to the one proposed by D. Collett (Modelling Survival Data in Medical Research – Section 3.6) with some appropriate alterations due to the fact that the interactions with the time variables were of main interest. All p-values throughout this procedure were calculated using likelihood ratio tests. At each step various criteria and statistics were examined (AIC, level-1 variance) to spot any unexpected changes. More importantly, since the coefficients of the two time-related variables cannot be judged independently (and thus the method of change in estimates cannot be used), we were producing graphs of the average BMI evolution over time for the three ART regimen groups (and of course keeping other variable at mean or baseline levels) to examine the trends for each group and their relation. It is noteworthy that the estimated difference in BMI between the highest (INSTI) and the lowest (NNRTI) groups at 4 years were very close to 1 Kg/m2 either using the final model presented in the manuscript or the full model (i.e. the one including all potential confounders as main effects and as interactions with the two time-related variables). Finally we would like to note that the method of change in estimates, although intuitive and widely used, has been recently criticized (see for example Talbot D et al. The change in estimate method for selecting confounders: A simulation study. Statistical Methods in Medical Research. 2021;30(9):2032-2044.). In any case, we added a sentence in the statistical methods subsection of the revised manuscript to provide more details regarding the model building procedure (page 6, end of 2nd paragraph)

3) Finally, I’m concerned about potential selection bias and generalizability. Only 63% of 5,323 ART initiators had viral load at 1 year. Only 49% of initial sample achieved viral suppression. Only 982 (18%) had complete baseline information. Does the proportion included in the final sample differ by regimen or calendar year. How representative are the 982 of the full sample?

We thank the reviewer for raising this interesting point. As we mention in the beginning of our results section and having checked for differences in sex, age, transmission category, AIDS at baseline, race and calendar year of ART initiation (and baseline BMI, CD4 and viral load wherever available), the only statistically significant differences are those reported in the manuscript. Thus the sample did not differ in terms of regimen class or calendar year and as we report the magnitude of the statistically significant differences (in e.g. sex or transmission category) were rather limited. We already mention this issue in the discussion section of our original version of the manuscript (“Finally, the exclusion of individuals ... are unlikely”).

Regarding the issue of viral suppression, the percentage of those with available measurements who achieved viral suppression was 76.7% which is reasonable for that period. However, we acknowledge that there is a serious problem with the availability of viral load measurements which was due to administrative/bureaucratic issues which affected all clinics and all PWH in Greece for a long period of time. We expanded the discussion on this issue in the revised version of our manuscript (page 24 , 2nd paragraph) and additionally we provide a table comparing the characteristics of those included and those excluded (due to missing baseline data) from our study, provided they were fulfilling the main criteria (starting boosted PI, NNRTI or INSTI based ART regimens, in 2010+, while ³18 years old, previously naïve with virologic response within the 1st year) as supplementary material.

Reviewer 2 Report

Peer review

Dear Editor,

Re: Changes in Body Mass Index after Initiation of Antiretroviral Treatment: Differences by Class of Core Drug

Thank you for inviting me to review the above manuscript. In their work, the Authors present the results of a Greek multi-cohort study exploring factors associated with BMI change in ART-naïve PWH initiating HIV treatment in 2010 or later and that achieved virological suppression within the first year of ART. Data were censored at 4 years after ART initiation, at the first ART switch or at the first virological failure, defined as two consecutive measurements >500 copies/mL after virological response.

Please find my observations below:

The manuscript is well structured, very well written and clear to read.

Statistical analysis: given the association between Black ethnicity, gender, and weight gain reported in the published literature, I believe that the inclusion of ethnicity in the models should be considered, albeit I appreciate the small numbers of non-White participants.

The excluded population is likely to represent an important source of bias, given the higher proportion of women, of people of ethnic minority background and with more advanced disease. It is important that adequate space is dedicated to reflect on this aspect in the Discussion. Perhaps a supplementary table should also be added to present all the data of this group (the excluded participants; identical to Table 1 in structure for the reader to have a look and compare). A flow-chart might help the reader in the understanding of the study population as well. In terms of missing data, multiple imputation might be used as a sensitivity analysis to try to overcome this issue. Could the Authors consider this approach to try to minimise the loss of data?

Table 1 should also include: the data on the backbone of the regimens; time on ART (in years); time since HIV diagnoses (in years).

I understand that a sensitivity analysis was conducted to factor in the NRTI backbone, given the association of TAF with weight gain. I am afraid this needs to be moved to the main analysis, as the interpretation of the role of ART in promoting weight gain cannot disregard the NRTIs that were used in the regimen. How about the use of TAF in NNRTI-based or PI-based regimens? Exploring the effect of TAF on INSTI-based regimens only appears arbitrary. I am afraid the main analysis will need to include the backbone for the manuscript to be considered for publication.

Author Response

Thank you for inviting me to review the above manuscript. In their work, the Authors present the results of a Greek multi-cohort study exploring factors associated with BMI change in ART-naïve PWH initiating HIV treatment in 2010 or later and that achieved virological suppression within the first year of ART. Data were censored at 4 years after ART initiation, at the first ART switch or at the first virological failure, defined as two consecutive measurements >500 copies/mL after virological response.

Please find my observations below:

1) The manuscript is well structured, very well written and clear to read.

We thank the reviewer for the overall positive assessment of our manuscript.

2) Statistical analysis: given the association between Black ethnicity, gender, and weight gain reported in the published literature, I believe that the inclusion of ethnicity in the models should be considered, albeit I appreciate the small numbers of non-White participants.

We agree with the reviewer that the assessment of any ethnicity effects on weight changes after ART initiation is important. Unfortunately, in our sample only 9 individuals were of black ethnicity (there were 11 more with the ethnicity variable missing but originating from African countries). We graphically explored BMI data of all these patients and there were only 2 cases with steep BMI increases: one 41 years old male increased his BMI from 17.3 to 22.8 Kg/m2 within 4 months and one 32 years old male increased his BMI from 22.9 to approximately 27 Kg/m2 within 18 months and remained stable thereafter. For the remaining black participants, BMI gains were much milder. Including ethnicity as a covariate in the final multivariable model revealed a slightly faster BMI increase but differences between black and non-black individuals in the rate of BMI change were not significant (p=0.718) as the width of the 95% CIs for black participants were almost 5 times wider compared to those for individuals of non-black ethnicity. Given these findings, we believe that the inclusion of ethnicity in our modelling efforts would not provide any reliable results thus we believe that it is better to just acknowledge that we don’t have enough data to investigate ethnicity effects.

3) The excluded population is likely to represent an important source of bias, given the higher proportion of women, of people of ethnic minority background and with more advanced disease. It is important that adequate space is dedicated to reflect on this aspect in the Discussion. Perhaps a supplementary table should also be added to present all the data of this group (the excluded participants; identical to Table 1 in structure for the reader to have a look and compare). A flow-chart might help the reader in the understanding of the study population as well. In terms of missing data, multiple imputation might be used as a sensitivity analysis to try to overcome this issue. Could the Authors consider this approach to try to minimise the loss of data?

We thank the reviewer for these insightful comments. In the revised version of our manuscript we provide a table (as supplementary material) which compares individuals fulfilling the basic inclusion criteria but excluded due to lack of baseline measurements to those included in the study sample. Additionally we provide a supplementary figure (flowchart) detailing all the steps of the sample selection procedure. We also discuss (page 24 , 2nd paragraph) in more detail the issue of missing viral load measurement which was caused mainly by administrative/bureaucratic issues and affected all PWH in Greece for a long period of time. Finally we extend our discussion on the potential generalizability issues due to the higher proportion of women and people with more advanced disease among those excluded due to lack of baseline measurements (page 24 , 2nd paragraph). Regarding the very reasonable suggestion for using multiple imputations, their application in the current setting would be very complicated and would require specialized software as the working dataset and the analyses are characterized by a hierarchical structure (repeated measurements on each individual) and a mixture of normally distributed continuous data, non-normally distributed continuous data and categorical data with two or more categories.

4) Table 1 should also include: the data on the backbone of the regimens; time on ART (in years); time since HIV diagnoses (in years).

Table 1 in the revised manuscript includes now information on ART backbone and time from HIV diagnosis to ART initiation. Since all individuals started ART while previously naive, Follow-up time in Table 1 corresponds to time on ART (subject to censoring – see end of 1st methods subsection)

5) I understand that a sensitivity analysis was conducted to factor in the NRTI backbone, given the association of TAF with weight gain. I am afraid this needs to be moved to the main analysis, as the interpretation of the role of ART in promoting weight gain cannot disregard the NRTIs that were used in the regimen. How about the use of TAF in NNRTI-based or PI-based regimens? Exploring the effect of TAF on INSTI-based regimens only appears arbitrary. I am afraid the main analysis will need to include the backbone for the manuscript to be considered for publication.

We understand the reviewers concerns regarding the role of NRTI backbone in the observed differences in terms of BMI rate of change between the 3 main classes of ART regimens. Probably it was unclear in the original version of the manuscript (as these data were not initially included in Table 1) but the role of TAF in non-INSTI regimens cannot be investigated in the current study as there were only 3 individuals with a TAF containing backbone and a NNRTI main drug (0.82%) while in the boosted PI group there were no individuals with a TAF containing backbone. In general there is very limited variation in backbone antiretrovirals in the boosted PI and NNRTI groups as the TDF+Emtricitabine combination is used in the vast majority of the cases (80.1% and 97.5%, respectively). As mentioned in the reply of the previous comment we now include backbone information in Table1 and we have also expanded the limitations section of our discussion (end of page 23) to acknowledge this weakness of our study.

Reviewer 3 Report

Overall I have enjoyed reading the manuscript which was clear. I would suggest the overview by a native speaker. the results are consistent with those from other studies.

From my point of view no major changes are needed.

Abstract: The sentence about "included individuals" and the last sentence of the abstract could be rephrased in order to be more clear.

Introduction: weight gain instead of weight gains

Methods: HIV-clinics in Greece (not HIV-1 clinics)

Viral rebound defined as two consecutive VL measurements>500c/ml: please confirm

Definitions: ..with 2 or more NRTIs ( probably not more than 2NRTIs were ever given)

Statistical analysis: CD4 cell count (not CD cell count)

Results: please rephrase the first sentence so it is more clear.

table1: would it be useful to add a column with overall results?

Discussion: the most clear part of the ms.

Define typical individuals

Explanation why PWH with AIDS before ART initiation gained more weight should be given

"Further clinical research is required ....": in this sentence i would advice to add with currently used ARVs

Limitations: a great proportion of the participants received old regimens (boosted Lopinavir and Atazanavir, Efavirenz). Female sex is overrepresented in the PI group. The short period of follow up evaluation for INSTIs is a major limitation.

Author Response

1) Overall I have enjoyed reading the manuscript which was clear. I would suggest the overview by a native speaker. the results are consistent with those from other studies.

From my point of view no major changes are needed.

We thank the reviewer for the overall positive evaluation of our work

2) Abstract: The sentence about "included individuals" and the last sentence of the abstract could be rephrased in order to be more clear.

We rephrased both sentences for more clarity.

3) Introduction: weight gain instead of weight gains

Fixed

4) Methods: HIV-clinics in Greece (not HIV-1 clinics)

Fixed

5) Viral rebound defined as two consecutive VL measurements>500c/ml: please confirm

Yes. This is indeed the definition we used. In the text we just clarify that the time of the rebound is considered to be at the first of these two measurements. Additionally, if the last available HIV-RNA measurement of an individual was >500 copies/ml we also censored follow-up at the time of this last measurement.

6) Definitions: ..with 2 or more NRTIs ( probably not more than 2NRTIs were ever given)

Indeed, there were no regimens with more than 2 NRTIs so we rephrased this definition.

7) Statistical analysis: CD4 cell count (not CD cell count)

Fixed

8) Results: please rephrase the first sentence so it is more clear.

This sentence was rephrased for more clarity. Additionally, a detailed flowchart for the study sample selection is now provided as supplementary material.

9) table1: would it be useful to add a column with overall results?

Column with overall results added.

10) Discussion: the most clear part of the ms.

We thank the reviewer for this positive comment

11) Define typical individuals

Done

12) Explanation why PWH with AIDS before ART initiation gained more weight should be given

Individuals with AIDS at baseline represent a relatively small (4.3%) fraction of the study sample thus the estimated differences, although statistically significant, are accompanied by large confidence intervals (now added in the revised version of the manuscript). These individuals start, as expected due to their poor health, from a lower BMI baseline but they also tend to gain weight faster, especially during the first 2 years of ART, most likely due to a “return to health effect” while some cases probably overcompensate for their bad pre-ART health status. The association between weight gain and low CD4 counts, which characterize individuals with AIDS, have been demonstrated in many studies We expand the discussion of this issue in the corresponding section of the revised manuscript (page 22, end of 2nd paragraph).

13) "Further clinical research is required ....": in this sentence i would advice to add with currently used ARVs

Done

14) Limitations: a great proportion of the participants received old regimens (boosted Lopinavir and Atazanavir, Efavirenz). Female sex is overrepresented in the PI group. The short period of follow up evaluation for INSTIs is a major limitation.

We now refer to the first two limitations (female sex in PI group and older ARVs) in the respective part of the discussion section of the revised manuscript (page 23, middle of last paragraph). Regarding the short follow-up in the INSTI group we were already listing it in the beginning of the limitations part of the discussion characterizing it as one of the “main” limitations but we also explain how we tried to overcome this issue without sacrificing too much sample size.

Round 2

Reviewer 1 Report

Thank for the additional reporting to address my concerns.